# What Causes Cancer in Women with a *gBRCA* Pathogenic Variant? Counselees’ Causal Attributions and Associations with Perceived Control

**DOI:** 10.3390/genes13081399

**Published:** 2022-08-06

**Authors:** Friederike Kendel, Katharina Klein, Stephen Schüürhuis, Laura Besch, Markus A. Feufel, Dorothee Speiser

**Affiliations:** 1Charité—Universitätsmedizin Berlin, Corporate Member of Freie Universität Berlin, Humboldt—Universität zu Berlin, and Berlin Institute of Health, Gender in Medicine, Charitéplatz 1, 10117 Berlin, Germany; 2Charité—Universitätsmedizin Berlin, Corporate Member of Freie Universität Berlin, Humboldt—Universität zu Berlin, and Berlin Institute of Health, Institute of Biometry and Clinical Epidemiology, Charitéplatz 1, 10177 Berlin, Germany; 3Division of Ergonomics, Department of Psychology and Ergonomics (IPA), Technische Universität Berlin, Marchstr. 23, 10587 Berlin, Germany; 4Charité—Universitätsmedizin Berlin, Corporate Member of Freie Universität Berlin, Humboldt—Universität zu Berlin, and Berlin Institute of Health, Hereditary Breast and Ovarian Cancer Center, Charitéplatz 1, 10117 Berlin, Germany

**Keywords:** hereditary breast and ovarian cancer, *gBRCA1/2-PV*, causal attributions, personal control, genetic counseling

## Abstract

Laypersons have a strong need to explain critical life events, such as the development of an illness. Expert explanations do not always match the beliefs of patients. We therefore assessed causal attributions made by women with a pathogenic germline variant in *BRCA1/2* (g*BRCA*1/2-PV), both with and without a cancer diagnosis. We assumed that attributions would be associated with the control experience. We conducted a cross-sectional study of N = 101 women with a g*BRCA*1/2-PV (mean age 43.3 ± 10.9). Women answered self-report questionnaires on perceived causes and control. Most women (97%) named *genes* as a causal factor for the development of cancer. Surprisingly, the majority of women also named *stress* and *health behavior* (both 81%), *environment* (80%), and *personality* (61%). Women with a cancer diagnosis tended to endorse more causes. The attributions to *personality* (ρ = 0.39, *p* < 0.01) *health behavior* (ρ = 0.44, *p* < 0.01), and *environment* (ρ = 0.22, *p* < 0.05) were significantly associated with *personal control*, whereas attribution to *genes* showed a small, albeit significant association with *treatment control* (ρ = 0.20, *p* < 0.05). Discussing causal beliefs in clinical counseling may provide a “window of opportunity” in which risk factors and health behaviors could be better addressed and individually targeted.

## 1. Introduction

When confronted with critical life events, people look for causes and explanation. This is especially true in the face of an illness. Understanding the event and being able to integrate it into one’s life is a strong need for patients. “Why me?” and “Why now?” are typical questions. There are rational as well as irrational explanations, secular as well as faith-based, medically informed as well as medically uninformed, and combinations of these different approaches.

Genetic research has shown that hereditary breast and ovarian cancer is caused by pathogenic germline variants. More specifically, inheritance follows an autosomal dominant pattern, it is independent of the sex of the inheriting parent, and it is of incomplete penetrance [1,2]. Women with a pathogenic germline variant in *BRCA1/2* (g*BRCA*1/2-PV) have, on average, an up to 72 percent risk of developing breast cancer during their lifetime [3]. Not every carrier of a g*BRCA*1/2-PV will develop the disease. In addition to genetic predisposition as the most important risk factor, other risk factors are discussed [4,5]. Factors such as age, family history, and health behaviors influence both the penetrance and pathology of hereditary carcinomas [6]. In their review, Daly et al. [7] compile evidence that modifiable risk factors, such as physical activity, lower body weight, and reduced alcohol consumption have a protective effect against cancer risk in mutation carriers.

Experts now believe that cancer is caused by a complex interaction of genetic and non-genetic factors [7,8]. However, empirical evidence is often inconsistent with the beliefs of affected individuals [9]. The systematic review by Dumalaon-Canaria, Hutchinson [9] on causal attributions showed that women with breast cancer subjectively considered family history, environment, stress and fate/chance as the main possible causes of their disease. Modifiable lifestyle factors played a rather minor role [9]. However, in a study in which the majority of women had a family history of cancer, the most frequently reported causal attributions were lifestyle, followed by heredity and environment [10].

Causal attributions are associated with health status [11] and psychological distress [12], and they also influence health behavior (e.g., [13,14,15]). Thus, understanding patients’ causal attributions is an important prerequisite for targeted counseling and therapy.

To date, there is a limited understanding of the causal beliefs of women with g*BRCA*1/2-PV. Therefore, the aim of this study was to describe the causal attributions made by women with g*BRCA*1/2-PV regarding the development of cancer or cancer recurrence. We explored whether women who had already received a cancer diagnosis and those who had not would differ in this regard. We also assumed a relationship between causal attributions and subjectively perceived control.

## 2. Materials and Methods

### 2.1. Procedures and Participants

This cross-sectional, observational, and mono-center study was conducted at the Center of Hereditary Breast and Ovarian Cancer at Charité—Universitätsmedizin Berlin, which belongs to the German Consortium of Hereditary Breast and Ovarian Cancer (GC-HBOC). Ethical approval was obtained from Charité—Universitätsmedizin Berlin (EA1/222/15). The study is described in more detail elsewhere [16,17].

Between August 2015 and April 2017, 300 women with a high statistical and/or familial breast cancer risk who had sought additional counseling were screened. Among them, 250 women were invited to participate in the study, and 207 women consented to participate. In total, 127 women returned the questionnaire. Of these, 101 participants met the inclusion criteria for the present data analysis: being of an age between 18 to 70 years and being a carrier of g*BRCA*1/2-PV. The exclusion criteria was insufficient German language skills.

### 2.2. Measurement

Sociodemographic and clinical variables were assessed with self-report questionnaires, supplemented by information from case report forms. To assess causal attributions, counselees were asked to rate six possible causes, namely *stress*, *genes*, *personality*, *health behavior*, *destiny*, and *environment* on a 4-point Likert scale ranging from 0 = “strongly disagree” to 3 = “strongly agree”. Additionally, counselees had the option to add two more possible causes and rate them on the same 4-point Likert scale. Two dimensions of the Brief Illness Perception Questionnaire (BIPQ) [18]—German version [19] were modified for use by women at high risk for breast and/or ovarian cancer to measure personal control (“How well do you think you can control your risk of breast and ovarian cancer with your health behavior?”) and treatment control (“How well do you think you can control your risk of breast and ovarian cancer with your prevention options?”).

### 2.3. Data Analysis

Relationships between causal attributions and personal control as well as treatment control, respectively, were calculated with Spearman’s Rho for ordinal variables. We conducted multiple linear regression analysis to determine the contribution of causal attributions (as predictors) to personal control (as dependent variable). The dependent variable in this model was an ordinal 10-point-scale, and was thus not perfectly applicable for a multiple linear regression model. However, diagnostic plots of the residuals of the model showed that the theoretical assumption to have independent and identically normally distributed residuals was approximately fulfilled.

All tests were performed using a two-tailed α level of 0.05. Data analysis was performed using IBM^®^ Statistical Package of the Social Science (SPSS)^®^ Statistics version 27 statistical software.

## 3. Results

### 3.1. Sample Characteristics

Table 1 provides an overview of demographic variables. Mean age of participants was 43.3 ± 10.9 years, 79.2% of women were partnered, 68.4% had children, 59.4% had more than 12 years of education, and 72.9% of the women were employed. Mean time since mutation test result was 14.2 ± 12.6 months, and 61.4% of the women had a g*BRCA*1-PV, and 38.6% a g*BRCA*2-PV. Among the participants, 52.5% of women had a cancer diagnosis, of which 83.3% had breast cancer and 9.4% had ovarian cancer, with a mean time since diagnosis of 62.1 months.

### 3.2. Causal Attributions and Control

Combining the two categories “agree” and “strongly agree”, it was found that most women (97%) named *genes* as a causal factor for the development of cancer, followed by *stress* and *health behavior* (both 81.2%), *environment* (80.2%), *personality* (61.4%), and *destiny* (36.6%). Causal attributions mentioned by the women in the free text fields included hormonal contraception (*n* = 1), side effect of tamoxifen (*n* = 1), pregnancy/breastfeeding (*n* = 1), and human papillomavirus (*n* = 1). In addition, women indicated causes that could be subsumed under *stress*, such as life circumstances or uncertainty (*n* = 3).

Figure 1 shows the distribution of the six main causal attributions for women with and without cancer diagnosis separately. There was no significant difference between the two groups. However, a trend emerged in that women with a cancer diagnosis expressed stronger agreement with all causes except environment.

### 3.3. The Association of Causal Attributions and Personal Control

Table 2 shows several significant correlations among *stress, environment, personality,* and *health behavior*. The largest correlation was found between *stress* and *personality* (ρ = 0.40, *p* < 0.01). The attribution to *genes* was not significantly associated with any of the other causal attributions, but showed a small, albeit significant, association with *treatment control* (ρ = 0.20, *p* < 0.05). *Personal control* was significantly associated with *personality* (ρ = 0.39, *p* < 0.01), *health behavior* (ρ = 0.44, *p* < 0.01), and *environment* (ρ = 0.22, *p* < 0.05). The correlation between *personal control* and *genes* was slightly, but non-significantly, negative (ρ = −0.12), indicating that if women identified genes as a possible cause, they reported less *personal control (*Table 2*)*.

In order to jointly model the extent to which *personal control* was independently associated with causal attributions, a multiple linear regression model was fitted with *personal control* as the criterion and causal attributions as predictors (Table 3). 

Table 3 displays the unstandardized regression parameter estimates ($\hat{\beta }$) together with their standard errors (*SE*), the 95% confidence intervals, the test statistic T, and the respective *p* values. The intercept value (4.026) represents the estimated constant regression term. *Personality* and *health behavior* were the most important and independent predictors of personal control. The six causal attributions together accounted for 27% of the variability of *personal control*.

## 4. Discussion

To our knowledge, this is the first study on causal beliefs among women with hereditary breast and ovarian cancer risk. We found that almost all participants believed that their genetic predisposition was causal in the development of cancer. However, four out of five women were also convinced that, in addition, *stress*, their own *health behavior*, and *environmental factors* could be significant causes. Two out of three women also named their *own personality* in this context. The belief that *genes* were a causal factor was associated with greater *treatment control*, i.e., the opinion that one could influence the development or recurrence of cancer with the preventive measures available. In contrast, the other causal attributions listed above strengthened women’s *personal control*, i.e., the belief that what matters is their own behavior.

Women diagnosed with g*BRCA*1/2-PV must deal with this finding for the rest of their lives. It is possible that causal attributions add a personal and existential meaning to their experience that goes far beyond the knowledge of “having a risk” [20]. That is, causal beliefs may help to integrate knowledge about one’s above-average risk into a broader narrative, an attempt to integrate what is happening into “a coherent life history” [21] (pp. 66–68).

g*BRCA* carriers seem to assume a multifactorial event, as do women without a hereditary risk. In our study, women who had already been diagnosed with cancer reported an even greater number of subjectively assessed causes than women without a cancer diagnosis. From a scientific perspective, the fact that not all women with g*BRCA*1/2-PV will develop breast or ovarian cancer during their lifetime argues for a multifactorial event. e.g., a woman with a g*BRCA*1-PV has an average lifetime risk of 72% of developing breast cancer [3]—but 28% of such women do not develop the disease. The factors involved in the development or recurrence of cancer, and the contribution of modifiable risk factors, are not yet completely understood. Intervention studies that focus on lifestyle factors have yielded encouraging initial results [22,23]. There is evidence that lifestyle (especially physical inactivity, obesity, alcohol, and smoking) may contribute to the development of cancer [5,7,8]. The same is true for environmental factors; for example, the negative influence of pesticides, traffic emissions, or radiation exposure [24]. This knowledge has now found entry into guidelines and is widespread in the population. Accordingly, the women in our study also felt that a healthy lifestyle made an important contribution to prevention.

### 4.1. Stress and Personality as Causal Factors

It was surprising that *stress* was mentioned so frequently as a causal factor in our study, as there is no empirical basis for this attribution (e.g., [9]). Nonetheless, in other studies, *stress* was also emphasized by cancer patients [9,25,26], and in some studies, breast cancer patients even considered it the central cause of their disease [27,28]. However, we had expected that the factor *stress* would be less important in a patient population for which the dominant causal factor—genetic predisposition—is known. Similarly surprising was the frequent mention of *personality* as a causal factor. Again, our results can be conceptually linked to other studies: The assumption of a so-called “cancer personality”, characterized in everyday language by “keeping everything inside” and “an inability to express negative feelings” such as frustration, fear, anger, or rage [29], persists among cancer patients, although there is no plausible explanation for such a connection [9]. Why are many women nevertheless convinced that stress or their individual personality exert significant influence on the occurrence of disease?

Those confronted with a serious illness look for interpretations of meaning, for causes to which accountability might be appropriate [30,31]. Taylor [32] describes how people seek not only explanations after a cancer diagnosis but also the biographical meaning that a cause may have for them. As the “why” is answered, a coherent narrative emerges that helps to reduce complexity [20]. To generate narratives, people often draw on information readily available to them, such as their own past experiences or anecdotal evidence. Many of the women with a positive gene test result were dealing with the potential of the disease in their family for some time, such as when a sister, aunt, or their own mother had the disease. Such prior experiences also influence causal beliefs, and these too can be included under the category of *stress* due to their threat potential.

In summary, *stress* is a broad construct under which very different events can be subsumed. It may refer to the illness of a relative, childhood trauma, divorce, economic worries, or recurring everyday stressors. Individual interviews would be needed to gain a more in-depth insight.

### 4.2. Fate as a Causal Factor

Only about one-third of the women in our study considered *fate* as a possible cause of their disease. Furthermore, in other studies [25,26], *fate* was mentioned less frequently than modifiable causalities. As in the case of *stress*, it is not clear what is meant by the term in each case. For religious patients, the disease may be seen either as a “punishment” from their God that does not allow healing, or as a “test” that encourages affected individuals to engage in positive coping behaviors. Secularly minded individuals may be more likely to associate the term “fate” with factors such as “chance” or “bad luck” [25,26]. Fateful events, in any case, are by definition beyond one’s control and cannot be influenced or controlled. In this respect, it fits into the picture that no association was found between *fate* and personal control in our study.

### 4.3. Personal Control

From clinical experience, we know that almost all patients raise the issue of non-genetic factors, such as *stress,* in the consultation. This usually reflects the hope of being able to influence what is happening. Alas, there is very little evidence to support this attribution. Should a doctor then discourage a patient’s belief due to ”a lack of evidence”? We advise to handle this sensitively, because, for one, stressors can indeed have indirect negative effects. For example, a person who continuously experiences everyday stress may take less time for physical activity or generally pay less attention to health, hygiene, and social contacts. Moreover, consistent with studies with other diseases and populations [33,34], our results show that causal attributions to behavioral and psychological factors are related to the experience of greater personal control. The expectation of being able to exert control over what happens could, in turn, have a preventive effect, in that patients may feel more confident in changing their health behavior. Such a relationship has been shown, for example, in a study of breast cancer patients in whom attribution to psychological factors increased the likelihood of lifestyle changes [35].

### 4.4. Study Limitations

Our study has several limitations. (1) These limitations include the relatively small sample. Our sample did not have sufficient power to detect differences in causal attributions among women with and without cancer. For example, to detect the difference in attribution to personality with a power of 80%, a sample size of *N* = 332 would have been necessary. One strength of our study, however, was that the sample, consisting of *gBRCA* carriers, was homogeneous in this respect. This provides insight into the causal attributions and associated factors in this specific population. (2) Our cross-sectional design does not allow causal interpretations. We cannot say with certainty whether causal attributions increase personal control, or by contrast, whether greater personal control leads to specific causal attributions. However, studies based on the model of Leventhal [36] using a longitudinal design tend to support the former interpretation [11,12]. (3) To date, there is no consensus on how causal attributions should best be recorded. Open-ended “most important cause” questions allow for individual recording, but have the disadvantage of making it difficult to summarize results, whereas closed-ended questions have the advantage of allowing different causal beliefs to be related to each other, rather than suggesting “the one important cause”. In our study, only six women mentioned a cause in the free text field other than the one given in the questionnaire, suggesting that the six categories we used were virtually exhaustive. (4) Since only women were included in our study, the results are not necessarily applicable to men. Studies with coronary heart disease patients, for example, have shown that men attribute more to *health behavior* and less to *stress* (e.g., [37]). 

### 4.5. Practical Implications for Counseling Women with gBRCA1/2-PV

The possibility of the onset of the disease and the uncertainty as to when it may occur or reoccur stimulates the need for subjective control. Affected women develop their own individual narratives, and it makes sense clinically and psychologically that these are acknowledged. In genetic counseling and care of high-risk patients, physicians can lay the foundations for managing uncertainty. Talking about causal beliefs may provide a “window of opportunity” in which the importance of risk factors and health behaviors can be better discussed and individually targeted. When appropriate, harmful misconceptions can be corrected. Similarly, when ideations are counterproductive, the physician can help provide relief and assurance that the patient is not to blame for the disease (“There’s no evidence of direct association”). Exploring causal beliefs allows for a deeper understanding of the patient’s experience, in whose experience the disease itself is a stressor [28], which can in turn be translated into more targeted counseling and therapies.

## 5. Conclusions

Most women with g*BRCA1/2*-PV consider a multitude of possible causes for the development of breast and ovarian cancer. Only a few of these factors have been empirically proven. Nevertheless, causal attributions are not “good” or “bad” per se. Health professionals should take their patients’ attributions seriously, and cautiously correct them where they are obviously counterproductive. Because some causal beliefs could be based on the hope that the disease can somehow be controlled, talking about attributions requires a high level of empathy. Due to the well-known relationship between attributions and health behavior, asking questions about causal beliefs may also provide an opportunity to openly discuss health behaviors that can (a) minimize the risk of disease and (b) improve quality of life in addition to clinical measures. In summary, our results suggest that patients’ beliefs have an important psychological function independent of their scientific evidence. In particular, beliefs that encourage and challenge patients to be active seem likely to enhance the health-promoting sense of personal control.

## Figures and Tables

**Figure 1 genes-13-01399-f001:**
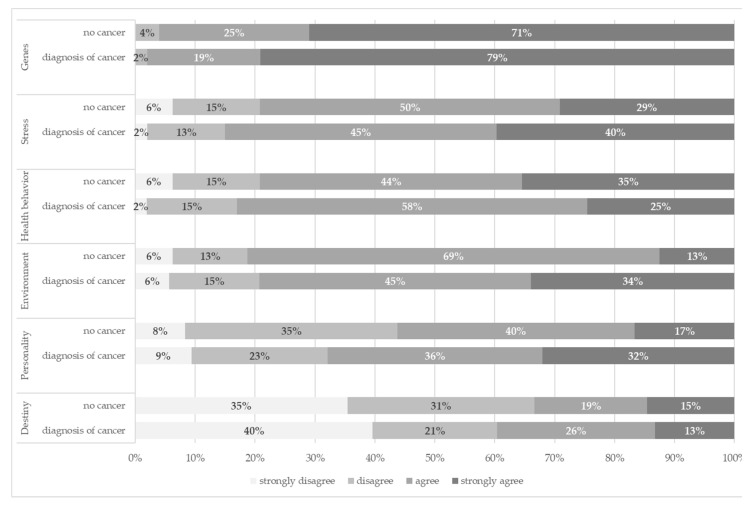
Distribution of causal attributions according to cancer diagnosis (*N* = 101).

**Table 1 genes-13-01399-t001:** Demographic and clinical characteristics of the entire study population and according to breast and/or ovarian cancer diagnosis.

	Entire Sample (*N* = 101)	Diagnosis of Breast/ Ovarian Cancer (*n* = 53)	No Breast/ Ovarian Cancer (*n* = 48)
Demographic characteristics			
Age (years), *M* (*SD*)	43.3 (10.9)	46.3 (9.8)	40.0 (11.2)
Partnership, *n* (%)			
Living with a partner	80 (79.2%)	38 (71.7%)	40 (83.3%)
Living without a partner	21 (20.8%)	13 (24.5%)	8 (16.7%)
Presence of children, *n* (%)	69 (68.4%)	37 (69.8%)	32 (66.7%)
Level of education, *n* (%)			
High school degree	60 (59.4%)	27 (50.9%)	33 (68.7%)
Secondary school	31 (30.7%)	17 (32.1%)	14 (29.2%)
Occupation status, *n* (%)			
Employed	72 (71.3%)	34 (64.2%)	38 (79.2%)
Unemployed	13 (12.9%)	7 (13.2%)	6 (12.6%)
Retired	14 (13.9%)	11 (20.8%)	3 (6.3%)
Clinical characteristics, *n* (%)			
Prophylactic surgery	46 (45.5%)	17 (32.1%)	29 (60.4%)
No Prophylactic surgery	13 (12.9%)	9 (17%)	4 (8.3%)
Prophylactic surgery	27 (26.7%)	13 (24.5%)	14 (29.2%)
Prophylactic salpingo-oophorectomy	11 (10.9%)	10 (18.9%)	1 (2.1%)
Mastectomy and salpingo-oophorectomy			
History of cancer, *n* (%)			
Breast cancer	-	44 (83.3%)	-
Ovarian Cancer	-	5 (9.4%)	-
Months since diagnosis, *M* (*SD*)	-	62.1 (62.5)	-
Pathogenic germline variant, *n* (%)			
*gBRCA1*	62 (61.4%)	35 (66%)	27 (56.3%)
*gBRCA2*	39 (38.6%)	18 (34%)	21 (43.8%)
Months since genetic analysis, *M* (*SD*)	14.2 (12.6)	14.3 (11.7)	14.1 (13.6)

**Table 2 genes-13-01399-t002:** Correlations between causal attributions and treatment/personal control (*N* = 101).

	Personal Control	Stress	Genes	Personality	Health Behavior	Destiny	Environment
**Treatment control**	---	−0.03	0.20 *	0.13	0.21 *	−0.01	<0.01
**Personal control**	---	0.22	−0.12	0.39 **	0.44 **	−0.02	0.22 **
**Stress**		1	−0.06	0.40 **	0.11 **	0.15 *	0.33 **
**Genes**			1	−0.01 ***	0.17 **	−0.01	−0.10 ***
**Personality**				1	0.36 **	0.14 *	0.33 **
**Health behavior**					1	−0.02	0.32 **
**Destiny**						1	0.22 **
**Environment**							1

*** *p* < 0.001; ** *p* < 0.01; * *p* < 0.05.

**Table 3 genes-13-01399-t003:** Regression of perceived personal control on causal attributions (*N* = 101).

Predictor	$\hat{\beta }$	*SE*	95%-*CI*	*T*	*p*
Intercept	4.026	1.379	[1.287, 6.764]	2.919	0.004
Stress	0.196	0.294	[−0.387, 0.780]	0.669	0.505
Genes	−0.725	0.419	[−1.556, 0.106]	−1.733	0.086
Personality	0.540	0.265	[0.013, 1.067]	2.034	0.045 *
Health behavior	1.062	0.316	[0.435, 1.689]	3.364	0.001 **
Destiny	−0.156	0.202	[−0.557, 0.245]	−0.772	0.442
Environment	0.086	0.313	[−0.536, 0.708]	0.275	0.784

** *p* < 0.01; * *p* < 0.05.

## Data Availability

The data that support the findings of this study are available on request from the corresponding author. The data are not publicly available due to ethical restrictions.

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
