# Peer review of "What Causes Cancer in Women with a gBRCA Pathogenic Variant? Counselees’ Causal Attributions and Associations with Perceived Control"

_genes, 2022, doi:10.3390/genes13081399_

Round 1

Reviewer 1 Report

While this study is well contructed, there are some limitations on how we can use these data in the clinical scenario. The women who self-selected participation by completing the questionnaire have already introduced a selection bias. This may be higher again when we consider that over 50% have a cancer diagnosis but are over 60 months (median) from their diagnosis. It may be that some of the "stress" highlighted byt he authors could be attribted to their cancer/treatment/life stressors but attributed for study purposes to the questions answered. 

This study is well written, but I do wonder how much this knowledge can translate in to clinical pratice, and thus how relevant theses findings may be to the journal readership. This should be explored in the conclusion

There are some small translation issues with the English (tabelle used repeatedly)

Author Response

Reviewer 1

While this study is well constructed, there are some limitations on how we can use these data in the clinical scenario. The women who self-selected participation by completing the questionnaire have already introduced a selection bias. This may be higher again when we consider that over 50% have a cancer diagnosis but are over 60 months (median) from their diagnosis.

We thank the reviewer for his or her comments. In fact, only slightly more than half of the women who met the inclusion criteria for this study completed the questionnaires. Such a rate has been reported by other studies and is, of course, a limitation. However, because the sample does not differ significantly from our patient population in the Familial Breast and Ovarian Cancer Center in terms of sociodemographic parameters, distribution of BRCA1 and BRCA2, and patients with and without a cancer diagnosis, we assume a largely representative sample. The major limitation is likely to be the sample size: the power was insufficient to detect differences between diseased and nondiseased women with respect to causal attributions. Another limitation is the cross-sectional design. We discuss these and other limitations transparently in the Limitations section.

It may be that some of the "stress" highlighted by the authors could be attributed to their cancer/treatment/life stressors but attributed for study purposes to the questions answered. 

If we understand the comment correctly, the reviewer means that the study questions themselves have an influence on the response behavior. Of course, this cannot be completely ruled out. We have therefore carefully compared the results with other studies: the results (especially the significant role of stress) can be very well classified here and also interpreted against the background of health theories.

This study is well written, but I do wonder how much this knowledge can translate in to clinical pratice, and thus how relevant theses findings may be to the journal readership. This should be explored in the conclusion.

We have gladly taken up this suggestion and have reviewed all the places where we have described the clinical implications. Among other things, we have added "Practical implications for counseling women with gBRCA" to the heading "Practical implications for counseling women with gBRCA" so that it is immediately clear to the reader how to apply this knowledge in their practice.

We also included the following sentences in the conclusion to emphasize the practical aspect of our findings:

“ (…) Health professionals should take their patients' attributions seriously and cautiously correct them where they are obviously counterproductive. Because some causal beliefs could be based on the hope that the disease can somehow be controlled, talking about attributions requires a high level of empathy. Due to the well-known relationship between attributions and health behavior, asking questions about causal beliefs may also provide an opportunity to openly discuss health behaviors that can a) minimize the risk of disease and b) improve quality of life in addition to clinical measures.” (…)

There are some small translation issues with the English (tabelle used repeatedly)

Thank you for mentioning! We have read the text again very carefully, and found and corrected several small mistakes.

Reviewer 2 Report

 I found this manuscript novel and presents valuable information about the impact of casual  beliefs on critical life events such as developing an illness. I found the introduction throughly covering the needed data, and this could be applied on the patients and methods section. In the results section, tables are adequately clarified .  Both dissuasion and conclusions are brief and presented focused details. 

The reference section are adequate. I finally recommend accepting the manuscript for publication. 

Author Response

Reviewer 2

I found this manuscript novel and presents valuable information about the impact of casual  beliefs on critical life events such as developing an illness. I found the introduction throughly covering the needed data, and this could be applied on the patients and methods section. In the results section, tables are adequately clarified .  Both dissuasion and conclusions are brief and presented focused details.

The reference section are adequate. I finally recommend accepting the manuscript for publication.

We thank both reviewers for their careful review and helpful comments.

In addition to the changes mentioned by the reviewers, we would like to make a small change in the title, which now reads: “What causes cancer in women with a gBRCA pathogenic variant? Intersection of counselees’ causal attributions and their associations with perceived control”